# Fractal Analysis of Tunnel Structural Damage Caused by High-Temperature and Explosion Impact

**Zhaopeng Yang [1] and Linbing Wang [1,2,*]**

1   National Center for Materials Service Safety, University of Science and Technology Beijing, Beijing 100083, China
2   The Sensing and Perception Lab., School of Environmental, Civil, Agricultural and Mechanical Engineering, University of Georgia, Athens, GA 30602, USA
*   Correspondence: linbing.wang@uga.edu

**Abstract:** The tunnel is one of the most important components in modern underground engineering. Due to long and narrow shape constraints, it very easily results in large-scale fire and explosion when deflagration is caused by the accidents of vehicles that transport dangerous goods in the tunnel. Previously, the studies on the damage to tunnel lining caused by high-temperature impacts in these kinds of disasters were often limited to a discussion of only one influencing factor, either fire or explosion, but they rarely considered the two factors simultaneously. In this work, the damage properties of full-size tunnel lining induced by high temperature and impact were evaluated, and the concrete samples from the whole lining arch were selected for CT scanning. The improved differential box-counting method was used for the fractal analysis of the CT images to obtain the damage-distribution properties of the tunnel lining structure under the two coupled influencing factors: the high temperature caused by fire, and the impact caused by deflagration.

**Keywords:** high temperature; concrete impact test; CT scan; fractal-dimension analysis; differential box-counting method

## 1. Introduction

Nowadays, more traffic lines are being transferred from the ground to the underground [1]. Although underground traffic tunnels bring great convenience to mankind, they inevitably have some potential safety hazards, such as higher casualties caused by the high-intensity explosions of oil tank trucks, and rescue difficulties due to tunnel collapse [2]. On the one hand, the disaster degree results from the high temperature and shock wave in cases of fire and explosion accidents in tunnels [3–5]. On the other hand, due to its relatively closed characteristics, the tunnel lining structure will be seriously damaged, and the collapse of the damaged lining will result in secondary damage to the accident site. The researcher should master the occurrence and development rules of secondary disasters caused by high-intensity explosions in tunnels, which is very important for the more scientific design of tunnel linings, avoiding disasters, and accurately improving the rescue process. Previous scholars' analyses on the damage mechanism of the tunnel lining structure have mainly focused on the two separate directions: single-combustion damage and explosion-shock-wave damage [6]. However, in actual accidents, high temperatures and explosion shock often occur together, and it is not enough to infer the damage results when considering only one influencing factor. In addition, due to the special shape of the tunnel itself, different flame heights and special phenomena, such as flame radiation and the plume impact on the ceiling, are significant. At the same time, most of the vehicle accidents will be close to the side of the pavement and will produce a special explosion. In this kind of explosion, there is a great difference between the side of the tunnel lining and the tunnel ceiling. From the perspective of practical engineering, it is necessary to consider the coupling effect of high temperature and the explosion impact on the damage of the

lining to make a more scientific design of the tunnel lining, which is useful as a reference in construction specifications.

At present, there are few studies on the combination of two influencing factors of tunnel lining accidents that focus on both high temperature and the explosion impact simultaneously. The design codes and standards are relatively simple, and they cannot evaluate the damage of the tunnel through the coupling of high temperature and shock impact. In this study, the damage analysis of a concrete tunnel lining structure was carried out according to the two influencing factors that are caused by the explosion of hazardous chemical vehicles after combustion in tunnels.

## 2. Experimental Procedure

### 2.1. The Fire-Damage Experiment of the Concrete Lining Structure

In cases of fire and explosion in tunnels, the temperature in the tunnel will rise instantaneously, and the tunnel lining structure will bear high temperatures. Fire and high temperatures will lead to the deterioration of the physical and mechanical properties of the lining materials and will damage the lining structure to varying degrees [7]. The structural stiffness of the lining material decreases due to the high temperature of fire, and the lining structure bearing normal loads will generate greater deformation. Due to the different temperature and impact strength at different positions on the same lining structure, the damage degrees are completely different at the mesostructure level.

This study aimed to simulate the fire situation, and to explore and study the damage law of the concrete lining structure at the mesostructure level. According to the Chinese national standard GB/T 22082-2017, eight precast full-size concrete lining segments were designed. The concrete material was C40 concrete, according to the national standard, shown in Figures 1 and 2.

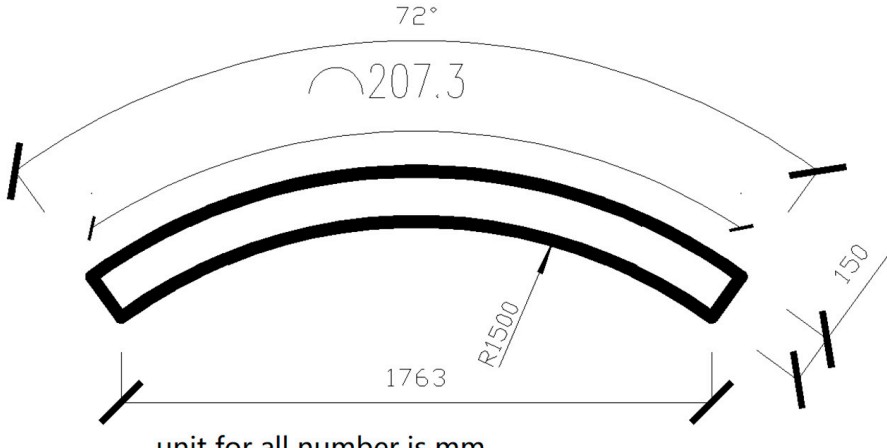

**Figure 1.** The detail size of the concrete lining structure.

Due to the long and narrow special structure of tunnels, the current research on the temperature field of tunnels mostly focuses on the high-temperature area of the arch crown and its longitudinal heat attenuation. However, there are few temperature tests on the vertical cross sections of the tunnel lining structure. For the core damage area, the uneven temperature on the cross section of the tunnel will cause high-temperature stress and eventually lead to structural instability and failure. In order to monitor the temperature distribution at different parts of the concrete lining arch, thermocouples were embedded in advance (shown in Figure 3). 4 pieces of full size concrete lining structure were hoist and sealed over the top of the open-hearth furnace (Figure 4). In this way, the temperatures of different parts and different levels of concrete can be monitored during the process of the combustion experiments.

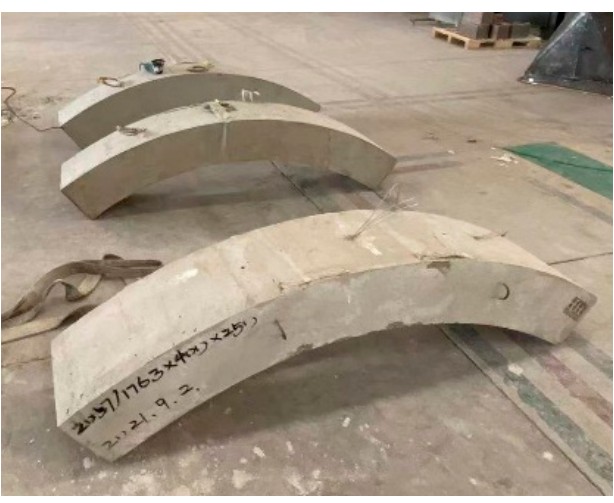

**Figure 2.** Concrete lining structure after demolding.

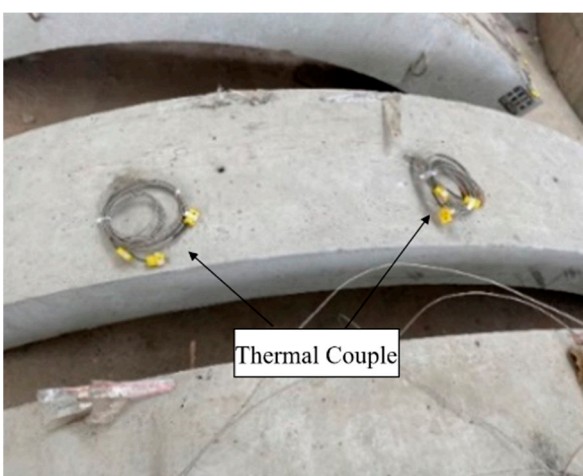

**Figure 3.** Embedded thermocouple at both 1/3 points of the arch.

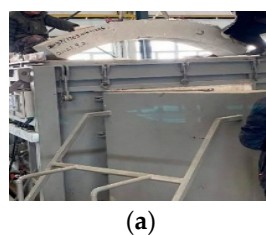

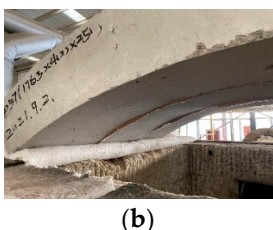

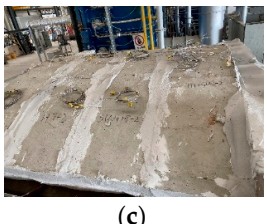

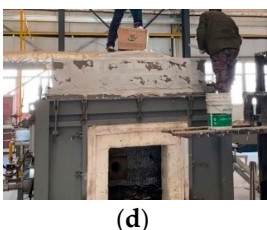

(**a**)            (**b**)            (**c**)            (**d**)

**Figure 4.** (**a**) Hoist the tunnel lining (**b**–**d**) and seal the lining structure over the high-temperature open-hearth furnace.

The heat source for this test was a high-temperature horizontal furnace that injects liquefied petroleum gas. In order to simulate fires under different fire-source scenarios in tunnels, different countries have adopted different standard temperature-rise curves. At present, the major international temperature-rise curves include ISO834, HC, RWS, and RABT. The temperature-rise curve of ISO834 is based on the burning rate of building materials, such as cellulose in the building-fire scene, which is inconsistent with the tunnel-fire scene. The HC curve is aimed at the petrochemical industry and offshore engineering. The RWS curve is aimed at the relatively closed-space environment, with less heat exchange with the surroundings, and so it is inconsistent with the semiclosed scene of a tunnel. The above heating curves have different degrees of limitations.

Compared with the above four kinds of curves, the RABT curve is developed based on a series of laboratory research results. In the first five minutes, the temperature increases rapidly to 1200 °C, and it is maintained from about 1 h to 1.5 h, and then the temperature starts to cool slowly. This curve is mainly used to simulate the hydrocarbon deflagration fire in the tunnel. When there is a lack of oxygen in the tunnel, the temperature begins to drop. Because its initial heating stage and cooling curve fully conform to the combustion characteristics of hazardous-chemical transport vehicles in semienclosed spaces, the combustion-damage test adopts the temperature-rise curve of RABT (GB/T 714-2007) (Figure 5). According to the heat-release rate and fan power of the furnace, a 3.3 h hydrocarbon curve (Type b) was chosen for this test, which also met the requirements of the temperature-rise curve of concrete lining structures to simulate the tunnel fire.

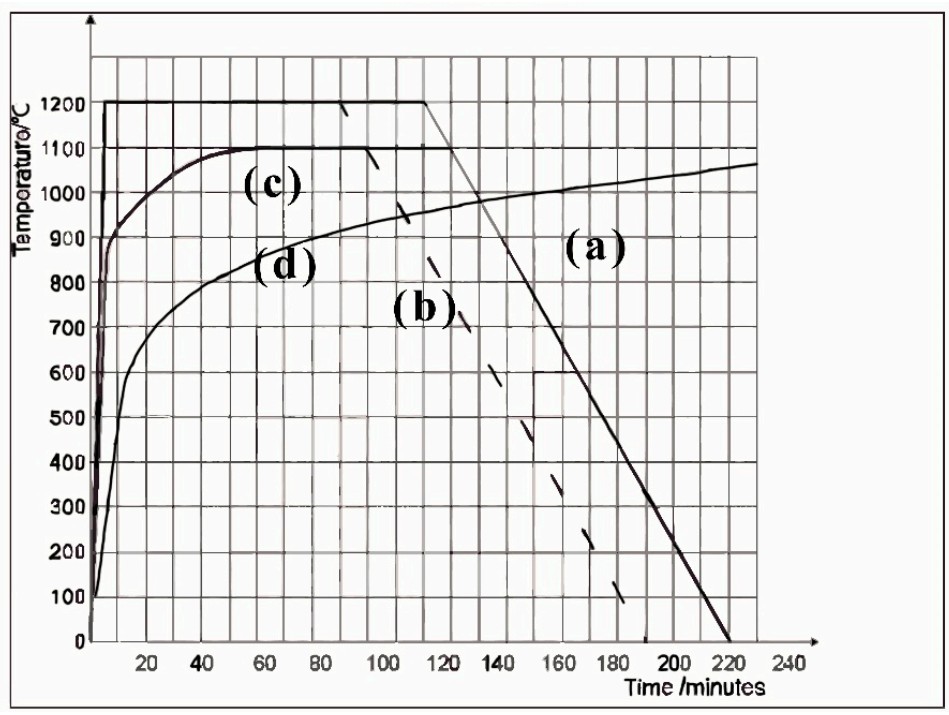

**Figure 5.** Four types of RABT curves: (**a**) 3.8 h for hydrocarbon fire; (**b**) 3.3 h for hydrocarbon fire; (**c**) 3.8 h for electric fire; (**d**) building-fiber fire.

Before the experiment, it is necessary to conduct a safety inspection on the high-temperature open-hearth furnace. Because the high-temperature effect of the hot-flue-gas flow on the tunnel lining structure was mainly considered, the Froude model was generally selected for the fluid judgment when analyzing the hot-flue-gas flow. This experiment focused on the combustion temperature-rise process, and so the flue-gas flow mainly depended on the high-power fan inside the furnace (Table 1).

**Table 1.** The detail process of the burning experiment of the concrete lining structure.

| | Process Name | Detail Procedures |
|---|---|---|
| 1 | Furnace Preparation | Check the pressure of liquefied petroleum gas (greater than 0.4 MPa), check the furnace thermocouple, and check the overall sealing of the furnace. |
| 2 | Concrete-Lining-Structure Hoisting Stage | The lining structure (four pieces as one group) is hoisted above the open-hearth furnace, and the gaps between the lining structures are sealed by high-temperature mastic to ensure that the heating rate in the furnace is consistent with the experimental requirements. |

**Table 1.** *Cont.*

| | Process Name | Detail Procedures |
|---|---|---|
| 3 | Pressure-Valve Check | After the valve test, upload the temperature curve (RABT) and select the combustion nozzle with an appropriate size (Burners 1 and 2 were selected for GB/t9978 curve; all nozzles were selected for GB/t26784). |
| 4 | Ignition Self-Inspection | After the self-inspection, manually start the fan and ignite the nozzle for the combustion test. |
| 5 | During the Burning | It is necessary to inspect whether the combustion state of each nozzle is normal, and to adjust the fan flow rate according to the real temperature-rise curve in the furnace to ensure that the temperature-rise curve meets the experimental requirements. |
| 6 | After the Experiment | The furnace door can only be opened after the cooling in the furnace is finished. |

After the burning experiment was complete, the temperature data inside the furnace were collected, and the temperature–time curve is shown in Figure 6.

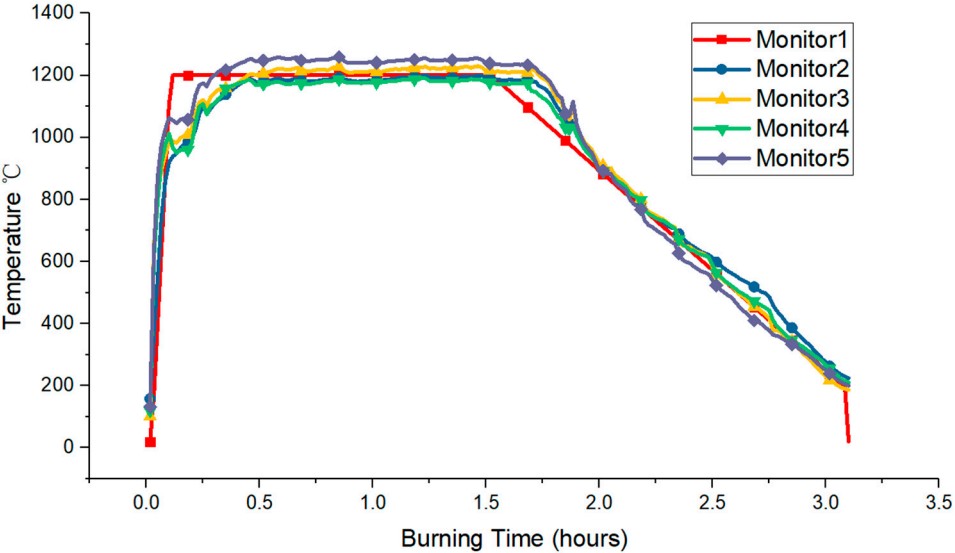

**Figure 6.** The temperature-rise curve inside of the furnace (RABT curve).

According to Figure 6, at 0–15 min, the test piece was in a significant heating period, and the temperature in the furnace quickly approached the peak value. Starting from 20 min, the temperature in the furnace gradually approached constant, and the peak value was maintained for nearly 1 h (1200 °C). From 1.75 h, the temperature in the furnace began to enter the cooling period, and there was almost no temperature difference at the several temperature-control points in the furnace, indicating that the combustion in the furnace was sufficient and the temperatures at different positions were very similar.

In this experiment, the thermocouples were embedded in the vault foot, arch foot, and arch center of the lining structure. Temperature data were collected by a multichannel acquisition instrument, and the location of the thermal sensor is shown in Figure 7. These data were helpful for us when analyzing the temperature of the critical area of the concrete lining arch structure.

In the fire scenario, the burning vault was impacted and wrapped by superheated air, and so its surface temperature was greatly affected by the flue-gas thermal convection. From Figure 8, it is clear that the surface temperature peak of the lining structure of the arch crown reached 900 °C, and there was also a high temperature of 680 °C in the concrete layer at the depth of 150 mm; however, due to the lack of superheated air at the arch feet on

both sides, the heat source often came from the heat radiation of the fire source itself, and so the corresponding temperature was only about half of that of the arch center. After the high-temperature open-hearth furnace heated the test piece according to the RABT curve, the arch center and arch foot of the test piece began to cool down gradually due to the thermal inertia of the concrete. Under the action of heat conduction, the temperature of the whole test piece began to concentrate to 400 °C; at 6 h, the temperature of the test block had dropped to about 300 °C. Due to the limitation of the furnace safety code, the concrete cannot be removed at high temperatures. Therefore, the drop-weight-impact test on the concrete lining structure could only be carried out after the cooling was completed.

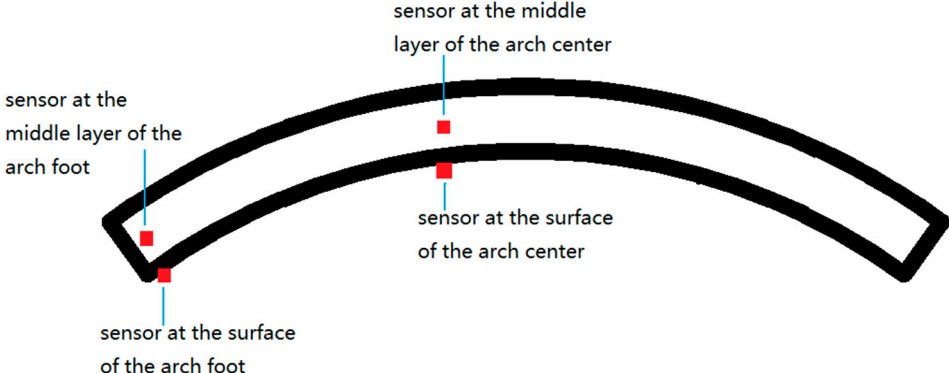

**Figure 7.** Location of thermal sensor.

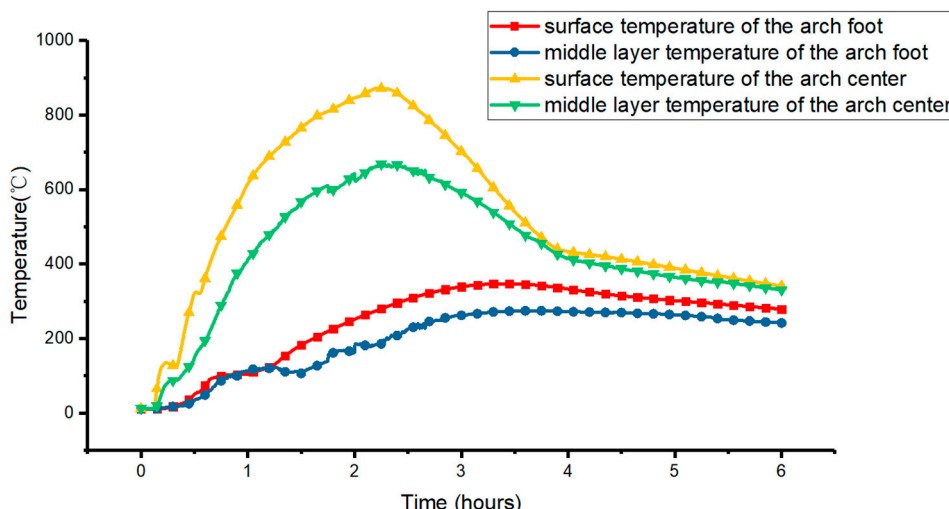

**Figure 8.** The surface and middle-layer temperatures of the center and foot of the concrete arch.

The surface color of the concrete structure changed due to the high temperature (Figure 9). The final color of the heating surface (880 °C) at the vault was grayish white and yellow, with cracks on the surface, and there was no pulverization (Figure 8). After the RABT-curve combustion, the thickness of the color change at the center of the arch was about 30 mm, and the surface also fell off during cutting. The rest of the concrete lining structures were ready for the drop-weight-impact test.

### 2.2. The Drop-Weight-Impact Test of Concrete Arch (Full-Size Tunnel Lining Structure)

So far, there is no standard for the impact-performance test of concrete [8]. According to the strain-rate range, the most commonly used experimental methods include the pendulum impact test, the MTS rapid loading test, the Hopkinson rod test, and the drop-weight-impact test [9]. Due to the limitation of the safety rules of the experimental site, the drop-hammer-impact test method was selected for this experiment. The gravity drop-weight test loads the test piece through the impact with the test piece after the heavy

hammer falls, and so it is suitable for high-strain-rate dynamic experiments, such as $10^{-1}{\sim}10^{-2}$ s$^{-1}$. Therefore, for the high-dynamic-loading experiment of the explosion impact of the tunnel lining structure, the drop-weight-test method was used in this study.

The drop-weight-impact test is the most practical experimental method, and it has the advantages of low requirements, low energy consumption, and simple evaluation. However, this method also has the disadvantages of poor controllability, poor bearing rigidity, and low accuracy. Moreover, the impact process of the drop-weight test is complex, and the extreme values of the impact force and impact loading times are generally affected by the drop mass, drop height, and overall stiffness of the test equipment. In order to avoid the above shortcomings, a totally new drop-weight-impact test frame was utilized to implement the high-precision measurement for these experiments. The drop-weight-impact frame is 3.7 m high, and the effective test height is 3 m. The maximum mass of the drop weight is 37 kg (Figure 10).

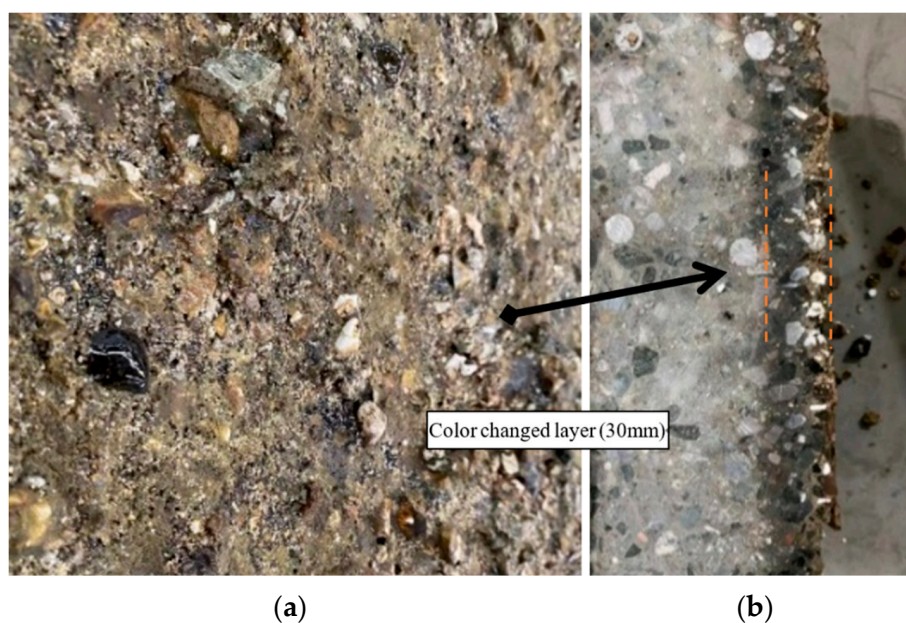

(**a**)      (**b**)

**Figure 9.** The surface (**a**) and cross section (**b**) of the concrete lining structure (central section).

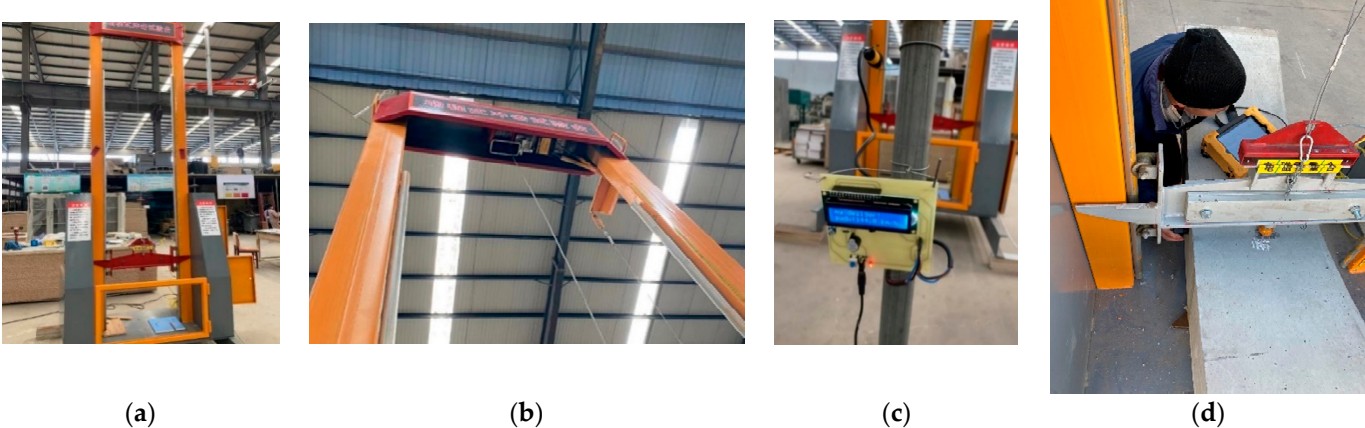

(**a**)      (**b**)      (**c**)      (**d**)

**Figure 10.** The drop-weight-impact system: the steel frame (**a**); the hoister (**b**); the drop-speed monitor (**c**); the drop weight (**d**).

In order to store more energy, two high-elastic-modulus springs were built into the left and right main columns of the test frame. When the falling weight is lifted, the

additional elastic potential energy can be accumulated by these two springs. In this way, the impact kinetic energy can be effectively increased through the conversion of the elastic potential energy at a limited laboratory height, which improved the feasibility for the indoor explosion-impact experiment. The counterweight can be adjusted to set the proper impact strength. From the perspective of safety, a locking device was installed to prevent unexpected dropping. Moreover, the control switch of the dropping weight was integrated on the movable platform.

Before the experiment, it is necessary to carry out the pre-experiment to set various impact conditions and record the experimental data of different heights. According to previous research, the peak value of the frontal impact pressure caused by the explosion from chemical transport vehicles in the tunnel was about 5~10 MPa [10,11]. Due to the limitations of the experimental conditions, the complete shock-wave form was not used in this study, but the highest wavefront pressure was matched.

The concrete block that we used in this experiment was a standard concrete test block poured in advance (150 mm × 150 mm × 150 mm, $f_{cu}$ = 30 Mpa). When repeating the experiment procedure, the impact-force sensor (Figure 11a) was installed under the concrete test block to obtain the time–history curve of the impact. In order to increase the overall stiffness of the concrete test block and provide the confining pressure, the concrete test blocks were surrounded by a 3 mm-thick steel plate (Figure 11b). The impact area can be obtained by measuring the depression area of the concrete test block. After measurement, the impact depression area was 0.05 m² (Figure 11c). The impact process was analyzed by the momentum impulse conservation. According to the speed of the falling hammer, the initial momentum of the falling hammer can be calculated. In the experiment, the loading frame was controlled by an electromagnetic switch and fell freely. After more than 2 m, the spring in the loading frame began to intervene and accumulate the elastic potential energy. Therefore, two different falling heights of 2 m and 3 m were selected to generate the different impact energies in this experiment. Before the impact started, the sensor was fixed on the base and an aluminum alloy plate was placed in the middle. After the drop weight hit the sensor, the load was collected by the dynamic-signal analyzer, and the data were recorded after all the experiments were finished (Table 2).

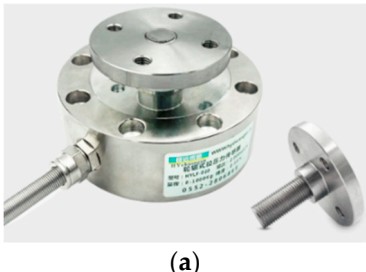 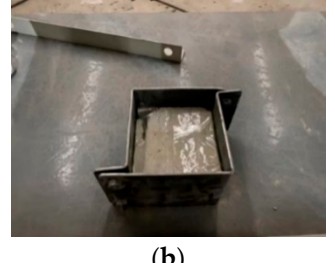 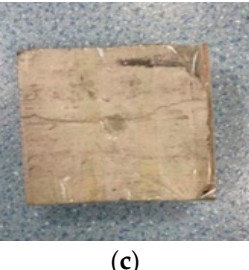

(**a**)  (**b**)  (**c**)

**Figure 11.** (**a**) Impact-load recorder; (**b**) concrete sample with restriction; (**c**) concrete sample after impact test.

**Table 2.** The impact-test results of unburned concrete block.

| Drop Mass (kg) | Drop Height (m) | Maximum Load (kN) | Standard Deviation of Maximum Load (kN) | P–T-Curve-Departure Dispersion (kN) | Final Speed (m/s) | Initial Momentum (kg·m/s) | Impulse (N·s) | Contact Time (ms) |
|---|---|---|---|---|---|---|---|---|
| 20 | 2 | 610 | 10.89 | 1.02 | 6.3 | 125.6 | 3050 | 5 |
| 20 | 3 | 630 | 2.12 | 0.24 | 9.7 | 193.6 | 3150 | 5 |
| 30 | 2 | 650 | 14.16 | 1.44 | 6.5 | 190.5 | 3250 | 5 |
| 30 | 3 | 690 | 17.05 | 1.71 | 10.2 | 306.2 | 3450 | 5 |
| 40 | 2 | 720 | 6.31 | 0.63 | 6.1 | 244.1 | 3600 | 5 |
| 40 | 3 | 790 | 12.4 | 0.85 | 9.5 | 380.1 | 3950 | 5 |

The initial momentum can be calculated according to the speed of the falling impact of the loading frame; the impulse can be calculated by integrating the time–history curve of the impact load (Figure 12). In the impact process, the momentum was less than the impulse, and so it could be inferred that, although the experimental frame with a rigid foundation was adopted, the loading frame had a certain degree of rebound in the impact process of up to 40 kg.

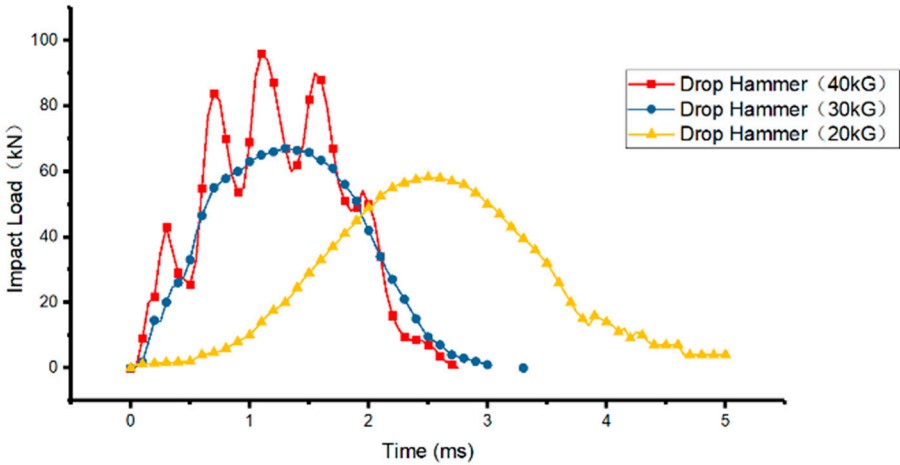

**Figure 12.** The impact-force–time curves.

It can be seen from Figure 11 that the pressure decreased rapidly after experiencing a main peak; as the distance from the center became greater, although the impact pressure experienced a main peak, the attenuation after the main peak was slower. The change in the overpressure indicates that the positive impact of the shock wave on the concrete was gradually weakened as the distance from the impact center became farther. The impact test in this chapter was only to debug the impact-loading frame and obtain the impact-load curve. We did not use it to carry out a damage analysis on the concrete samples in this chapter. The real 2D damage analysis will be carried out in the next chapter.

### 3. The Result and Discussion with Fractal Analysis

*3.1. The Fractal Dimension and Concrete-Mix Proportioning*

The fractal dimension of concrete is a quantitative index that describes the geometric properties of concrete, and so some material parameters or physical quantities related to the geometric properties of concrete can be quantitatively expressed as a function of the fractal dimension. The fractal dimension is directly related to the geometry and geometric distribution.

The particle size and distribution of concrete aggregate have a great impact on the properties of concrete, not only directly affecting its mechanical properties, such as stiffness, strength, and durability, but also the mix proportion, hydration heat, dry shrinkage, creep, and other properties of concrete. Because the aggregate itself is not greatly deformed during concrete pouring and molding, the space between aggregates is only filled with smaller aggregates and cement mortar.

When research regards the aggregate material in concrete as one set, the fractal dimension is basically only related to the distribution of the aggregate, which is often referred to as the gradation.

Fractal geometry is used to describe the irregularity of complex bodies. Its main feature is that, when describing the form in nature, it is no longer an integral dimension but a fractional dimension in space [12].

Scale invariance means that the geometric figure does not have the corresponding geometric characteristics at a specific scale. When the scale changes, the geometric characteristics of the object do not change. Therefore, for the graphics with fractal characteristics, the geometric characteristics of some areas are always consistent with the original graphics

after being enlarged and reduced at the same scale. The spatial distribution and geometric shape of the fine structure contained in the structure do not change. The functional expression of self-similarity and the scale invariance of the graphics is Equation (1):

$$F(\varepsilon\delta_0) = \varepsilon^{-D}F(\delta_0) \tag{1}$$

where the function ($F(\varepsilon\delta_0)$) is the characteristic function of the set, $\delta_0$ is the expanded scale, and $\varepsilon$ is the proportional relationship between different function mappings.

There is fractal effect in the aggregate set of concrete, and this fractal effect is one of its geometric-distribution characteristics. Therefore, the fractal effect is the cumulative distribution form of all the elements in the set, which is a functional map with the fractal effect. According to Fuller's point of view, the distribution of aggregate follows Equation (2):

$$p = \left(\frac{x_i}{x_{\max}}\right)^{\eta} \tag{2}$$

where $\eta$ is the Fuller grading index, and $x_i$ is the size of the aggregates.

However, the Fuller grading function is a discrete function, and the function of the local fractal is a continuous differentiable function. The fractal dimension can be used to reflect the structural properties of concrete at the mesoscale.

The larger the fractal dimension of the cross-section image of concrete, the more complex the graph is, and the closer it is to the full graded aggregate set. According to the Fuller formula, when $\eta$ is closer to 2, this means that the aggregate in the image is closer to the set with full grading; if $\eta$ is less than 2, this indicates that the concrete aggregate is closer to the single gradation.

It is assumed that the aggregate size ($x$) obeys the normal distribution, which has the mean value ($x_i$), and the variance is $\sigma = 2$. The probability density function can be generalized and integrated to obtain the average value of the aggregate particle size by Equation (3):

$$\overline{x}^{+} = \frac{1}{\sqrt{2\pi}\sigma}\int_{\mu}^{+\infty} xe^{-\frac{(x-\mu)^2}{2\sigma^2}}dx = \mu + \frac{\sigma}{2\sqrt{2\pi}}$$
$$\overline{x}^{-} = \frac{1}{\sqrt{2\pi}\sigma}\int_{-\infty}^{+\mu} xe^{-\frac{(x-\mu)^2}{2\sigma^2}}dx = \mu - \frac{\sigma}{2\sqrt{2\pi}} \tag{3}$$

where $\overline{x}$ is the average aggregate size; $\overline{x}^{+}$ is the aggregate size larger than the average ($i + 1$ interval); $\overline{x}^{-}$ is the aggregate size smaller than the average.

The average aggregate size is also related to the number of aggregates of each interval. The average aggregate size ($\overline{x_i}$) of Interval $i$ can be written as Equation (4):

$$\overline{x_i} = \frac{N_i^{(1)}(x_i)\overline{x_i}^{+} + N_i^{(1)}(x_i)\overline{x_{i-1}}^{-}}{N_i^{(1)}(x_i) + N_i^{(1)}(x_i)}, 1 < i < n \tag{4}$$

where $N_i^{(1)}$ is the number of aggregates at the size interval ($x_{i-1}$, $x_{i+1}$). In the differential interval $[x, x + dx]$, the differential increment ($dN$) can be expressed as Equation (5):

$$dN(x) = -D\frac{M_0}{\rho Kv}x_{\max}^{D-2}x^{-1-D}dx \tag{5}$$

where $k$ is the common ratio of each aggregate size interval after forming an equal ratio series. After integrating both sides of the above formula, the number of aggregates in any aggregate size range ($[x_1, x_2]$) can be obtained as Equation (6):

$$N(x)|_{x_1}^{x_2} = \int_{x_1}^{x_2} -D\frac{M_0}{\rho vK}x_{\max}^{D-2}x^{-1-D}dx = \frac{M_0}{\rho vK}x_{\max}^{D-2}(x_1^{-D} - x_2^{-D}) \tag{6}$$

where $N(x)$ is the number of aggregates, and $D$ is the fractal dimension of the image. Therefore, we can deduce that, in the CT scanning image, the fractal dimension of the aggregate itself will not change, and it is equal to 2. However, considering that cement mortar itself also has a fractal effect, the fractal of the concrete CT scanning image should be a positive real number less than 2.

### 3.2. The Box-Counting Method

The traditional box-dimension method uses the minimum number of boxes that can fully cover the edge of the fractal image as a factor in the analysis of the fractal dimension of the image. However, now, the difference-box-dimension method is more competent in engineering. The box-covering method used by Mandelbrot analyzes and estimates the fractal dimension of the British coastline, but the fractal dimension obtained by this method only reflects its overall shape, and the damage in concrete is more uncertain.

For the idea and method of the differential box dimension for an image with a given area of m × m, it is assumed that it has been decomposed into small blocks, and S is an integer between 1 and m/2. At this time, the image of m × m can be regarded as an image between the second dimension and third dimension ((x, y)), and it can be regarded as the coordinate position of the pixel in the plane; the gray value of the pixel can be regarded as the third dimension. In this way, the image of m × m can be regarded as a series of small boxes with a volume of ($s * s * \widehat{s}$) on each grid. Assuming that the values with the highest gray value and lowest gray value in the image fall in the $l$ and $k$ small boxes, respectively, the distribution of the set in the $(i, j)$ grid is Equation (7):

$$n_r(i,j) = l - k + 1 \qquad (7)$$

### 3.3. Fractal Analysis of Unburned Concrete after Impact Test

For the unburned concrete lining structure, after impact, a water mill was used to cut the concrete lining structure according to a cross-sectional area of $100 \times 100$ mm (Figure 13). After the cutting, the test block was selected for CT scanning. From the scanned images, the vertical-section images of the concrete test block were selected as representative of the damage image.

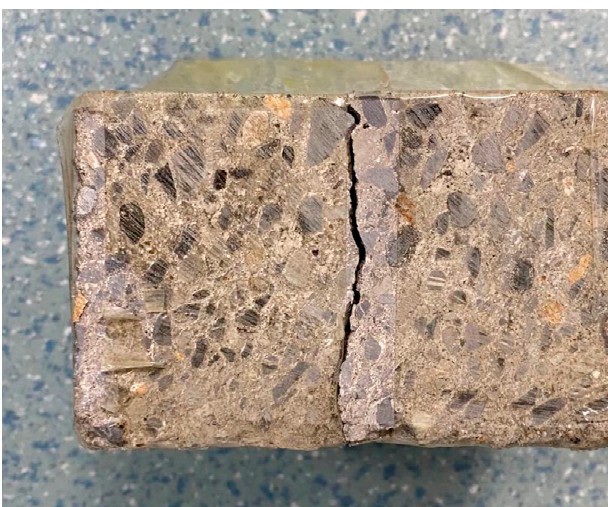

**Figure 13.** The unburned concrete sample after the impact test (3 m high; drop mass: 40 kg).

CT scanning was performed for the unburned concrete test block with a size of $150 \times 150 \times 150$mm. The thickness of CT scanning is 1 mm. After the initial scanning, 120 CT images were obtained for each concrete test block. The obtained CT images were extracted and analyzed by the MATLAB program. MATLAB provided the imread function to record the RGB value of the pixels in the image. Next, the obtained RGB images were

transformed into the gray images by the rgb2gray function. There are many methods of gray processing. The algorithm of the weighted average of the RGB value was adopted in MATLAB. After the image was binarized, the image edge (edge function) was obtained by the edge function (Figure 14).

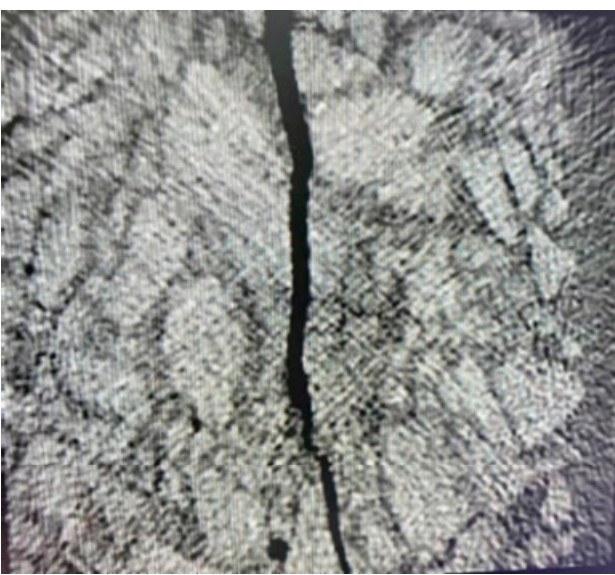

**Figure 14.** The CT scanning images of the middle part of the unburned concrete standard test block after impacting ($150 \times 150$ mm size).

The following step outlines the damage boundary in the concrete-section picture. The CT-scan gray image ($M \times M$) was divided into a grid ($s \times s$, $1 < s \leq \frac{M}{2}$, s = positive integer). After obtaining the size of the image, the appropriate box size must be selected. Calculating the scale divided the image into boxes and calculated the value of each box at the same time. Finally, the number of boxes under this scale in the image area was obtained by Equation (8):

$$N_r = \left( \sum_{i,j} n_r(i,j) \right) \Big/ S^2 \tag{8}$$

where $S^2$ is the area of the chosen box, and $N_r$ is the total number of the boxes that cover the crack in the picture. For the change in r caused by different box sizes (s), the least-squares method can be used to fit it. The vertical axis is $\log N(r)$, where $N(r)$ is the number of boxes; the horizontal axis is $\log(\frac{1}{r})$, where r is the ratio of the side length of the box and this gray image. The slope of the fitting curve is the corresponding fractal dimension (D) (Figure 15).

A total of 20 concrete test blocks in four batches were subjected to the impact tests at different heights, and the images in the same layer were collected for the fractal-dimension fitting. The results are shown in Table 3.

It can be seen from Table 3 that, with the increase in the loading quality, the concrete materials gradually entered the damage state from the initial state. The whole test blocks gradually expanded from a small number of cracks to local cracks in the region, and the crack-development route was relatively single. With the continuous increase in the loading quality, more cracks gradually appeared along the impact direction, and the cracks gradually became thicker and wider.

From the perspective of the fractal dimension, when the test block was not impacted, the fractal dimension of the test block remained at about 1.593–1.607. When the impact mass increased from 20 kg to 30 kg, the fractal dimension increased from about 1.621–1.639 to 1.649–1.672. When the impact mass finally reached 40 kg, the fractal dimension increased to 1.723–1.784.

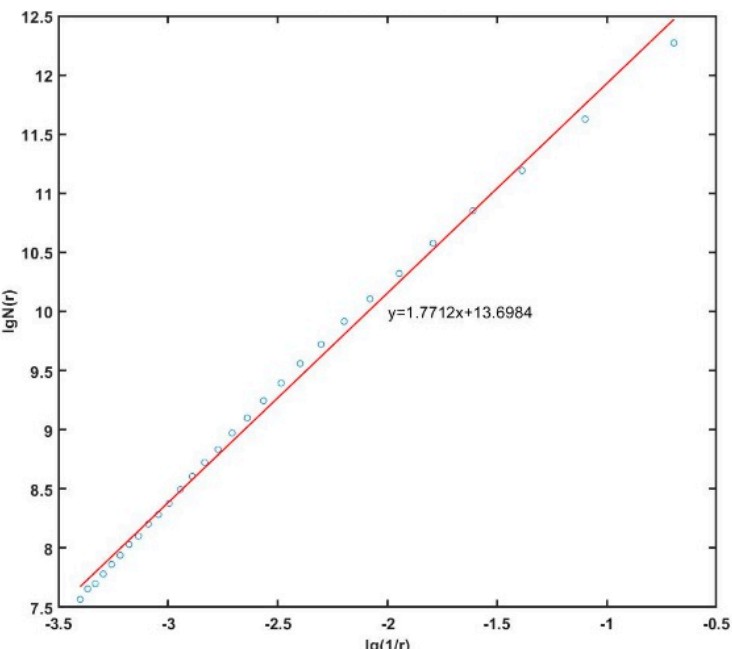

**Figure 15.** The fractal-dimension fitting curve of the unburned concrete sample.

**Table 3.** The fractal dimensions of all unburned concrete samples.

| No Impact | Sample 1 | Sample 2 | Sample 3 | Sample 4 | Sample 5 |
|---|---|---|---|---|---|
| | 1.593 | 1.609 | 1.582 | 1.596 | 1.607 |
| 20 kg (3 m-height impact) | Sample 6 | Sample 7 | Sample 8 | Sample 9 | Sample 10 |
| | 1.637 | 1.621 | 1.639 | 1.634 | 1.629 |
| 30 kg (3 m-height impact) | Sample 11 | Sample 12 | Sample 13 | Sample 14 | Sample 15 |
| | 1.661 | 1.672 | 1.654 | 1.664 | 1.649 |
| 40 kg (3 m-height impact) | Sample 16 | Sample 17 | Sample 18 | Sample 19 | Sample 20 |
| | 1.772 | 1.786 | 1.723 | 1.769 | 1.784 |

*3.4. Analysis of Different Parts of the Concrete Lining Structure with Both Burning and Drop-Weight-Impact Tests*

In order to make a comparison with the traditional damage variables, the data processing of this study adopted the range-normalization-change theory, and the concrete CT image was expressed by the damage variables based on the fractal dimension. The range-specification change refers to finding the maximum and minimum values for each variable in the data matrix. The variation between the maximum value and minimum value is the range. Then, the minimum value is subtracted from each variable and is divided by the range to obtain the normalized data.

It can be seen from the impact images of the vault that the direction of the impact crack of 20 kg (Figure 16) is relatively single, multiple cracks longitudinally paralleled to the impact direction are distributed in the middle of the sample, and there are basically no transverse cracks. In the 30 kg impact image (Figure 17), there are more bifurcations at the upper and lower ends of the main crack. The 40 kg impact has an obvious transverse crack (Figure 18). In addition to the longitudinal transverse main crack, the small gap between aggregate and aggregate is distributed on the left and right sides of the sample, indicating that the shock wave caused visible damage to the interface transition zone of the aggregate and aggregate in this sample.

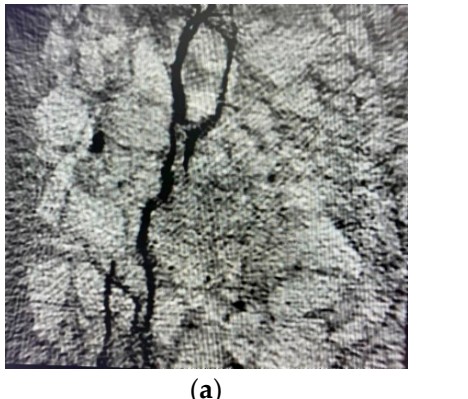 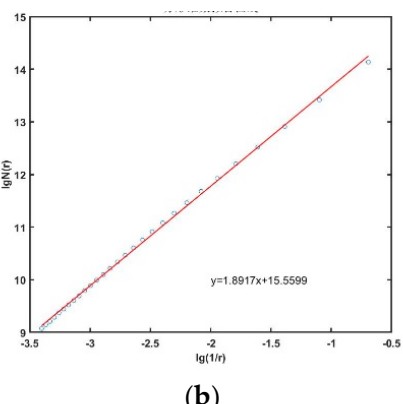

| (**a**) | (**b**) |

**Figure 16.** The damage picture of the center of the arch by impact (20 kg) and burning (**a**); the fractual curve (**b**).

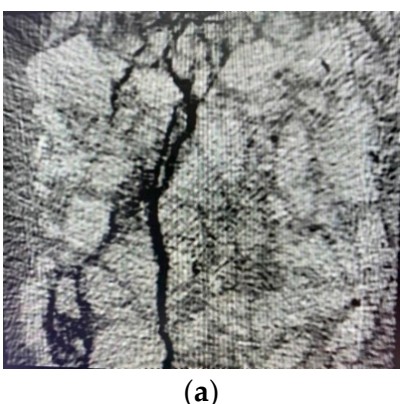 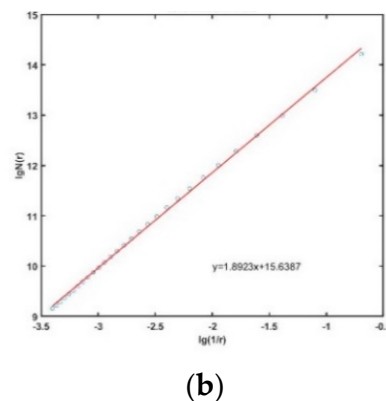

| (**a**) | (**b**) |

**Figure 17.** The damage picture of the center of the arch by impact (30 kg) and burning (**a**); the fractual curve (**b**).

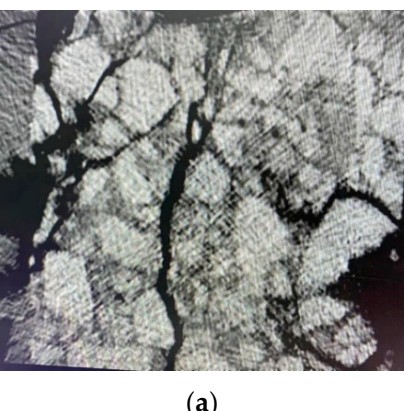 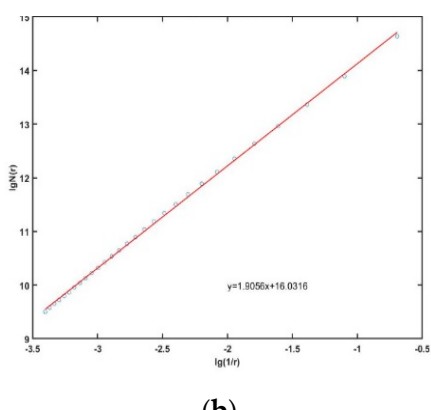

| (**a**) | (**b**) |

**Figure 18.** The damage picture of the center of the arch by impact (40 kg) and burning (**a**); the fractual curve (**b**).

Due to the deviation of the samples, the arch center may not be able to express the damage of the tunnel lining structure in the mesodimension. Therefore, in this experiment, the samples at the arch crown and arch foot were selected for the CT scanning. Finally, the damage situations of each distribution point under different temperatures and different impact conditions were obtained, and they are shown in Tables 4 and 5.

**Table 4.** The fractal dimensions of concrete samples after burning (center of the arch).

| No impact | Sample 21 | Sample 22 | Sample 23 | Sample 24 | Sample 25 |
|---|---|---|---|---|---|
| | 1.602 | 1.608 | 1.592 | 1.598 | 1.605 |
| 20 kg (3 m-height impact) | Sample 26 | Sample 27 | Sample 28 | Sample 29 | Sample30 |
| | 1.726 | 1.746 | 1.735 | 1.743 | 1.743 |
| 30 kg (3 m-height impact) | Sample 31 | Sample 32 | Sample 33 | Sample 34 | Sample35 |
| | 1.806 | 1.816 | 1.804 | 1.817 | 1.793 |
| 40 kg (3 m-height impact) | Sample 36 | Sample 37 | Sample 38 | Sample 39 | Sample40 |
| | 1.903 | 1.893 | 1.907 | 1.892 | 1.913 |

**Table 5.** The fractal dimensions of concrete samples after burning (foot of the arch).

| No impact | Sample 41 | Sample 42 | Sample 43 | Sample 44 | Sample 45 |
|---|---|---|---|---|---|
| | 1.591 | 1.583 | 1.597 | 1.591 | 1.601 |
| 20 kg (3 m-height impact) | Sample 46 | Sample 47 | Sample 48 | Sample 49 | Sample 50 |
| | 1.632 | 1.646 | 1.635 | 1.613 | 1.649 |
| 30 kg (3 m-height impact) | Sample 51 | Sample 52 | Sample 53 | Sample 54 | Sample 55 |
| | 1.716 | 1.723 | 1.716 | 1.721 | 1.736 |
| 40 kg (3 m-height impact) | Sample 56 | Sample 57 | Sample 58 | Sample 59 | Sample 60 |
| | 1.772 | 1.781 | 1.763 | 1.774 | 1.778 |

In the damage process, the growth of the microdefects of the material itself will eventually lead to the failure of the material. In this study, the data were processed by range normalization. Range normalization was used to find the maximum and minimum values of each variable in the data matrix. The deviation between these two values was the range, and the minimum value from each variable was subtracted and divided by the range to obtain the normalized data with Equations (9) and (10):

$$A_{ij}^* = \frac{A_{ij} - \min(A_{ij})}{R_j} (i, j = 1, 2, 3 \ldots n), 0 \leq A_{ij}^* \leq 1 \tag{9}$$

$$R_j = \max(A_{ij}) - \min(A_{ij}) \tag{10}$$

After the specification changed, each variable of the data became the data with the minimum value of 0 and the maximum value of 1.

In this study, the minimum value of the fractal dimension is 1.582 of the No. 3 sample without combustion and impact, and the maximum value is 1.913 (inner layer: 600 °C) of the No. 5 sample with a 40 kg impact after combustion, and so the range value is 0.331. Therefore, similar to Table 4, it can be obtained as shown from Tables 6–8.

These data showed that the fractal dimension of cracks can be used as an index to predict the safety performance of concrete lining structures. To further analyze the relationship between the temperature and fractal dimensions, the samples were set randomly in five groups.

The results shown in Figure 19 indicate that the fractal dimension was not linear proportional to the temperature. In the process of increasing the temperature, the fractal dimension became lower at a temperature around 300 °C, which may have been caused by the increase in small cracks that may homogenize the concrete geometry at the mesoscale. After the temperature continuously increased to 650 °C, the cracks became more complex and obvious, which the fractal dimensions increased to a higher level.

**Table 6.** The normalized shock-damage data of unburned concrete samples.

| No Impact | Sample 1 | Sample 2 | Sample 3 | Sample 4 | Sample 5 |
|---|---|---|---|---|---|
| | 0.033 | 0.082 | 0.051 | 0.042 | 0.076 |
| 20 kg (3 m-height impact) | Sample 6 | Sample 7 | Sample 8 | Sample 9 | Sample10 |
| | 0.166 | 0.118 | 0.172 | 0.157 | 0.142 |
| 30 kg (3 m-height impact) | Sample11 | Sample12 | Sample13 | Sample14 | Sample15 |
| | 0.239 | 0.272 | 0.218 | 0.248 | 0.202 |
| 40 kg (3 m-height impact) | Sample16 | Sample17 | Sample18 | Sample19 | Sample20 |
| | 0.574 | 0.616 | 0.426 | 0.565 | 0.61 |
| Temperature-Peak Range of Samples 1–20 | Room temperature (20–22 °C) | | | | |

**Table 7.** The normalized shock-damage data of concrete samples after burning (foot of the arch).

| No Impact | Sample 21 | Sample 22 | Sample 23 | Sample 24 | Sample 25 |
|---|---|---|---|---|---|
| | 0.027 | 0.03 | 0.045 | 0.027 | 0.057 |
| 20 kg (3 m-height impact) | Sample 26 | Sample 27 | Sample 28 | Sample 29 | Sample 30 |
| | 0.151 | 0.193 | 0.16 | 0.094 | 0.202 |
| 30 kg (3 m-height impact) | Sample 31 | Sample 32 | Sample 33 | Sample 34 | Sample 35 |
| | 0.405 | 0.426 | 0.405 | 0.42 | 0.465 |
| 40 kg (3 m-height impact) | Sample 36 | Sample 37 | Sample 38 | Sample 39 | Sample 40 |
| | 0.574 | 0.601 | 0.547 | 0.58 | 0.592 |
| Temperature-Peak Range of Samples 41–60 | 261–274 °C | | | | |

**Table 8.** The normalized shock-damage data of concrete samples after burning (center of the arch).

| No Impact | Sample 41 | Sample 42 | Sample 43 | Sample 44 | Sample 45 |
|---|---|---|---|---|---|
| | 0.06 | 0.079 | 0.046 | 0.048 | 0.069 |
| 20 kg (3 m-height impact) | Sample 46 | Sample 47 | Sample 48 | Sample 49 | Sample 50 |
| | 0.435 | 0.495 | 0.462 | 0.486 | 0.486 |
| 30 kg (3 m-height impact) | Sample 51 | Sample 52 | Sample 53 | Sample 54 | Sample 55 |
| | 0.677 | 0.707 | 0.671 | 0.71 | 0.637 |
| 40 kg (3 m-height impact) | Sample 56 | Sample 57 | Sample 58 | Sample 59 | Sample 60 |
| | 0.97 | 0.94 | 0.982 | 0.937 | 1 |
| Temperature-Peak Range of Samples 21–40 (arch center) | 642–664 °C | | | | |

The results showed that the fractal dimension increased with the load, which can better characterize the roughness of the fracture section. To find out which was the dominated condition for the damage of the concrete samples, five groups of samples were set with the same load but different temperatures. These curves proved that there is a strong functional relationship between the fractal dimension and temperature. To demonstrate the diverted damage degree between different groups, it is necessary to use Equations (9) and (10) to convert the fractal dimension into equivalent damage degrees.

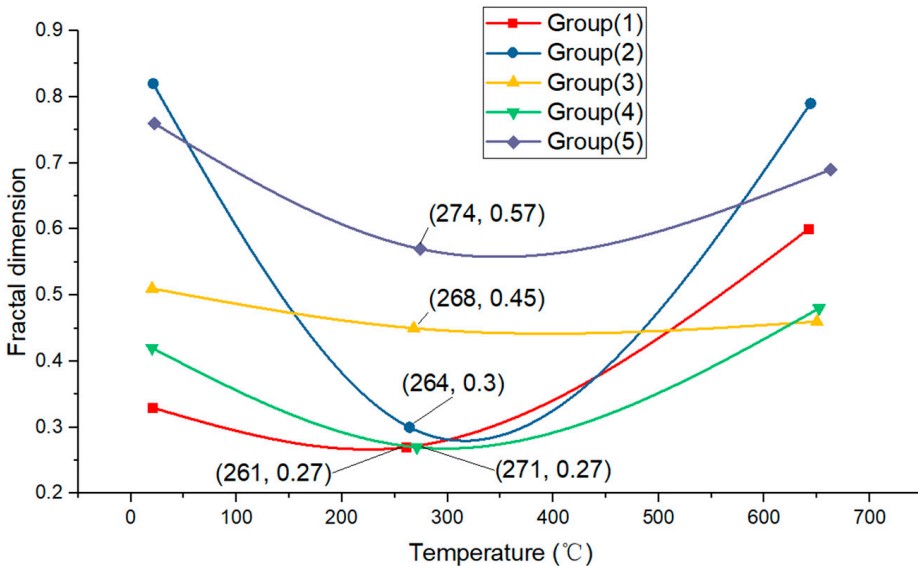

**Figure 19.** Fractal dimensions of different groups of samples at different temperatures.

From Figure 20, it can be seen that the unburned samples had nearly the same damage degrees as the samples from the foot of the arch (0.094–0.202). The reason for this is that the temperature (around 265 °C) is not high enough to trigger the full dehydration of C-S-H in concrete material. A 20 kg impact apparently did not induce too much damage to the structure of the foot of the arch; when the temperature increased to 650 °C (at the center of the arch), it was clear that even with the same 20 kg impact, the damage degree increased significantly (around 0.7). This means that when the shock wave is attenuated in the tunnel (away from the explosion over 15 m), most of the damage area will be the celling of the tunnel, as this area has the highest temperature.

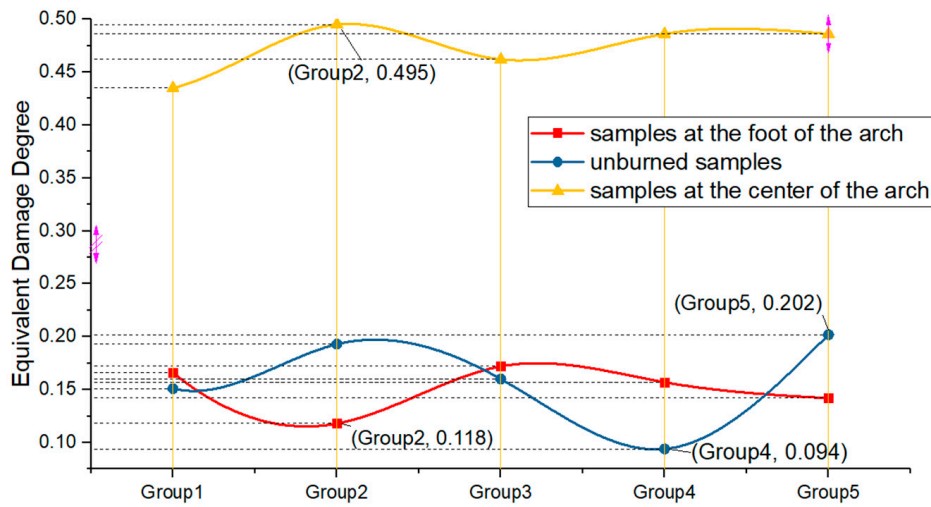

**Figure 20.** The 20 KG impact test for different parts of concrete lining structure.

When the drop weight increased to 30 KG (Figure 21), all the samples increased their damage degrees, and the samples at the foot of the arch had a higher amount of damage than the unburned samples. When the drop weight reached 40 KG (Figure 22), the maximum damage showed up and the damage degree was merely the same as at the foot of the arch and for the unburned sample.

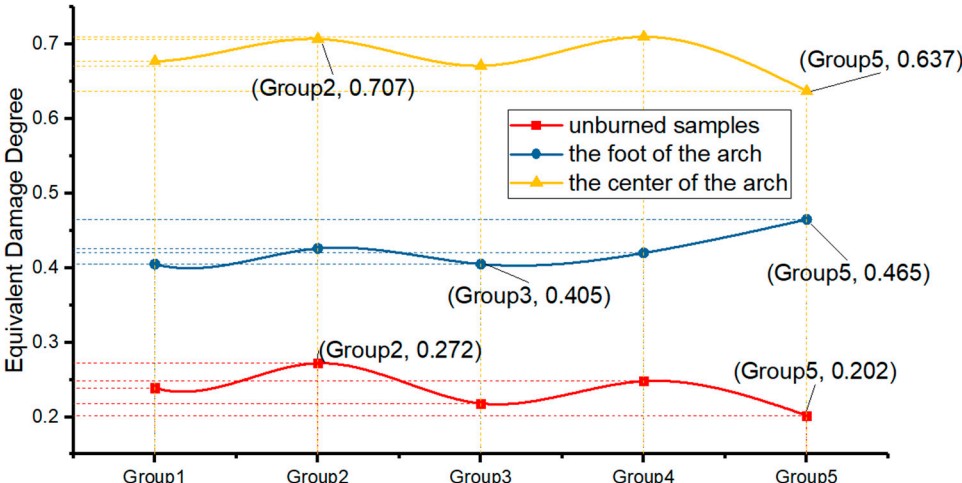

**Figure 21.** The 30 KG impact test for different parts of concrete lining structure.

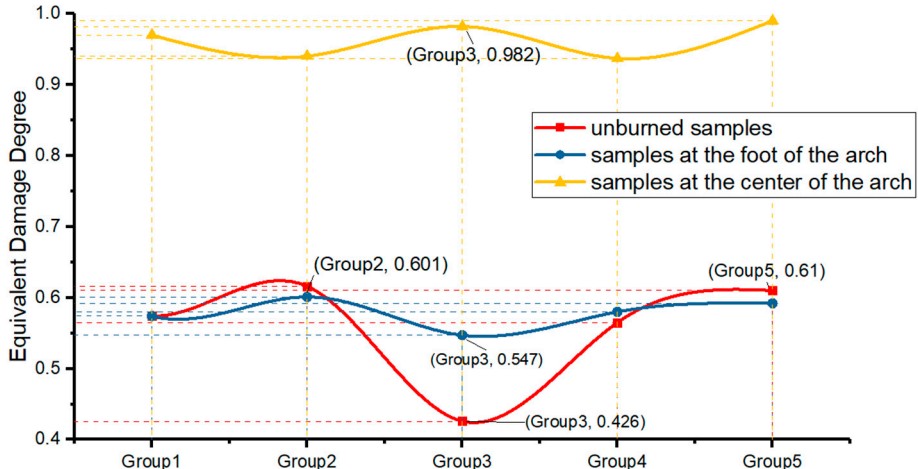

**Figure 22.** The 40 KG impact test for different parts of concrete lining structure.

### 4. Conclusions

This research has successfully quantified the fire- and blast-damage analysis for the concrete lining structure based on the fractal-dimension theory. To achieve this, the proper scale needs to be carried out. The scale here has two meanings. The first refers to the size selection of the concrete CT images; that is, the corresponding relationship between the fractal dimension and temperature-impact damage under different scales was obtained through concrete CT images of different sizes. X-ray CT scanning can visually and qualitatively capture the development path of internal mesocracks. The length, width, curvature, and density of microcracks can be measured by statistical methods, but these are not suitable for large-scale analysis because of their low efficiency. The microcracks in concrete have self-similarity, and the region is a multifractal body. Because of its accurate coverage and intelligent calculation, fractal-geometry theory is an effective means to analyze the self-similarity of concrete materials.

The second is to adjust the size of the box and the area covered by the box so as to further subdivide the damage descriptions of concrete under different scales. The damage of concrete under impact and high temperatures has obvious fractal characteristics. At the initial stage of loading, the fractal dimension increases slightly with the increase in the loading, and it increases significantly with different temperatures.

Conclusion 1: In the process of concrete damage progression, with the increase in stress, the damage variable will also increase. However, under the small equivalent impact

load, the concrete will experience the process of compaction, and the damage variable calculated according to the fractal dimension will be reduced. If the equivalent of the impact load is large enough, then the crack in the concrete increases obviously, and the capacity of the whole test block expands.

Conclusion 2: The influence of temperature on the test block is also extremely obvious. Because the maximum surface-temperature difference between the arch crown and arch foot is about 500 °C, and the temperature difference in the layer is about 320 °C, the damage fractals after the same impact experience are obviously different. From the results of the unburned test blocks, the impact damage of 20 kg was about 0.15, that of 30 kg only increased to 0.23, and that of 40 kg reached 0.56. The damage of the tunnel lining structure caused by impact alone is far less than that caused by high temperature plus impact. For the arch foot after the RABT combustion curve, the damage at a 20 kg impact was 0.16, and the impact at 40 kg was 0.57. It can be seen that after 300 °C and 2 h of combustion, the impact resistance of the concrete components did not decrease significantly. However, in the vault-damage experiment, the impact of 20 kg was 0.47, and the peak value reached 0.95 when it reached 40 kg.

Conclusion 3: The dynamic strength of concrete materials will be significantly stronger than the static strength. This enhancement effect is generally described by the dynamic increase factors (DIFs) or the stress–strain-curve relationship. According to the literature, most of the compressive-strength-enhancement effect of concrete under a high strain rate comes from its own structural effect. Obviously, the increase in temperature inflicts important damage on the aggregate structure of concrete. Therefore, when the concrete structure is subjected to high temperatures and impact simultaneously, the high-temperature-damage proportion of concrete will be more serious. Therefore, considering that dangerous vehicles explode and burn in tunnels, further study should focus on strengthening the thermal insulation for tunnel lining structures.

**Author Contributions:** Conceptualization, Z.Y. and L.W.; methodology, Z.Y.; software, Z.Y.; validation, Z.Y. and L.W.; formal analysis, Z.Y.; investigation, Z.Y.; resources, Z.Y.; data curation, Z.Y.; writing—original draft preparation, Z.Y.; writing—review and editing, L.W.; visualization, Z.Y.; supervision, L.W.; project administration, L.W.; funding acquisition, L.W. All authors have read and agreed to the published version of the manuscript.

**Funding:** This research received no external funding.

**Institutional Review Board Statement:** Not applicable.

**Informed Consent Statement:** Informed consent was obtained from all subjects involved in the study.

**Data Availability Statement:** Data available on request from the authors.

**Conflicts of Interest:** We declare that we do not have any commercial or associative interest that represents a conflict of interest in connection with the work submitted.

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
