# Peer review of "Fractal Analysis of Tunnel Structural Damage Caused by High-Temperature and Explosion Impact"

_buildings, doi:10.3390/buildings12091410_

Round 1

Reviewer 1 Report

The manuscript reports a systematic study on the material degradation of concrete used in tunnels subject to fire and impact. A series of real size specimen were designed and tested with comprehensive results obtained, which were thoroughly analyzed and studied. The results show that high temperature can cause significant material degradation with considerable increase in fractal numbers. The outcome of the study is valuable and can be of interest to many colleagues in the field. The reviewer is happy to recommend it for publication subject to following revisions.

(1) Add unit dimensions in Figure 1

(2) Show the accurate positions of thermal couples in Figure 3.

(3) Delete Figure 6.

(4) Add Figure 8.

(5) Show the positions of the 4 thermal sensors in Figure 7.

(6) Delete Section 4.1 two repetitive paragraphs.

(7) Delete the Chinese words in Figure 14.

Reviewer 2 Report

Fractal analysis of tunnel structural damage caused by high-temperature and explosion impact

Zhaopeng Yang1, Linbing Wang

Is this being what you consider as the contribution and novelty of this paper?

There is no clear structure of the paper where we can distinguish between the testing methodology and the results & discussion! You are mixing up all together. Please separate the section of testing methodology from that of results & discussion.

You are abusing with the use of “we”! In technical writing we use the third person.  

The joint between Concrete lining structure pieces (arches) will have an effect over the all behavior of the assembled structure. In real work, the number of these joints are limited. How do you assess the joints effect? You need some discussion on this issue.

 Figure 5:

Is there any practical reason or explanation why the temperature of around 1200 C is maintained for about 1.75 hours?

Also, why the ascending curve is sharp while the descending one (cooling) is slow??

Mostly there is too much description and very little discussion of the results. The authors are required to justify, explain, discuss their results not only report and describe.

Reviewer 3 Report

This research has successfully quantify the fire and blast damage analysis based on the fractal dimension theory. The topic is nice and the content is going to be useful. The experiment work and analysis content are sufficient. The reviewer has the following queries which need to be addressed in revising the paper:

1. In this paper, two kinds of disasters, fire and explosion, were considered on tunnel structures. However, actually in this study, the impact tests were conducted after the tunnel structures were cooled down. That actually is the dynamic properties or impact properties after fire, not the expression “simultaneously” in the paper. After all, there do exist some researches in the existing publications that conduct impact tests on structure when it is under fire. So it is suggested to modify related expressions to make a distinction.

2. For the fire test, in consideration of that no load was considered during this test, so the main conclusion of this part is the temperature distribution inner the tunnel structures when given a surrounding high temperature. It is suggested to mainly focus on this point, to figure out the temperature distribution law maybe.

3. In the impact part, it is believed that the used steel plate is to generate a confined pressure, so it can represent a 3D stress state. My question is what is the quantitative relation between the real condition of tunnel structures and the steel-plate-confined concrete block? If this question is not clear, then what is the relation of the impact test part with the tunnel structures? Please explain.

4. In the impact part, the information about concrete block is too rare, and that makes this part hard to understand. Please make a modification.
